# Missense Mutations in the KAT Domain of *CREBBP* Gene in Patients with Follicular Lymphoma: Implications for Differential Diagnosis and Prognosis

**DOI:** 10.3390/ijms26146913

**Published:** 2025-07-18

**Authors:** Anna Smolianinova, Ivan Bolshakov, Yulia Sidorova, Alla Kovrigina, Tatiana Obukhova, Nelli Gabeeva, Eduard Gemdzhian, Elena Nikulina, Bella Biderman, Nataliya Severina, Nataliya Risinskaya, Andrey Sudarikov, Eugeniy Zvonkov, Elena Parovichnikova

**Affiliations:** National Medical Research Center for Hematology, 125167 Moscow, Russiadusha@blood.ru (A.S.);

**Keywords:** follicular lymphoma, *CREBBP*, NGS, chemotherapy

## Abstract

Follicular lymphoma (FL) is one of the most common types of non-Hodgkin’s lymphomas. The tumor is characterized by a wide range of clinical manifestations, ranging from indolent forms to early transformation and progression with a poor prognosis. The search for clinically significant genetic changes is essential for personalized risk assessment and treatment selection. The *CREBBP* gene is frequently mutated in this type of lymphoma, with changes occurring at the level of the earliest tumor precursor cells. However, the prognostic and diagnostic significance of the *CREBBP* gene mutation status in FL has not been fully established. In this study, we analyzed sequencing data of exons 22–30 of the *CREBBP* gene in 86 samples from patients with different grades of FL (1–3B), including those in the 3A–3B subgroup without the t(14;18) translocation. We also investigated the prognostic significance of *CREBBP* gene mutations in relation to the treatment options, namely high-dose chemotherapy with autologous hematopoietic stem cell transplantation (HDCT/auto-HSCT) and conventional chemotherapy programs (CCT). It was found that FL patients with a single missense mutation in the KAT domain of the *CREBBP* gene experienced an extremely low number of early adverse events related to lymphoma and had better long-term survival rates, regardless of treatment option. In contrast, when comparing patients with FL without a missense mutation in the KAT domain or those with multiple mutations in the *CREBBP* gene, overall and progression free survival were worse, and early progression and histological transformation were more common. Compared to standard therapy, patients who underwent HDCT/auto-HSCT in the FL 1–3B (14;18)-positive group without a single missense mutation in the KAT domain had better survival rates and lower rates of transformation and early progression. In addition, among patients with FL 3A–3B (14;18)-negative, we found that there were no cases of a missense mutation in the KAT domain of the *CREBBP* gene. This suggests that a single missense mutation in the *CREBBP* gene may be a feature that discriminates 14;18-positive FL with a favorable prognosis from a high-risk disease. FL 3A–3B (14;18)-negative may represent a distinct variant with different biology and underlying mechanisms of development compared to classical FL.

## 1. Introduction

Follicular lymphoma (FL) is the most common type of indolent lymphoma in adults, accounting for approximately 2.2–5 cases per 100,000 people. According to the World Health Organization classification, FL is a type of tumor that originates from B-lymphocytes of the germinal centers, with at least some follicular growth patterns [1]. In fact, such a broad definition encompasses a wide range of diverse morphological, immunophenotypic, genetic, and clinical variants of the disease, which contribute to the heterogeneity of the disease.

First and foremost, the tumor exhibits clinical heterogeneity. Most FL patients are diagnosed at an advanced stage (III/IV), and only a small percentage of patients (10–15%) have a localized form of the disease. In 45–65% of cases, patients with stages I-II of FL can be successfully cured with local interventions such as radiation therapy or surgical resection of the affected lymph nodes [2]. Despite its sensitivity to modern immunochemotherapy, the advanced stage of the disease is characterized by multiple relapses and is considered incurable. Approximately 20% of patients with advanced FL develop atypically early relapse after first-line therapy, within 24 months of treatment initiation (POD24). Until recently, the unfavorable prognosis was thought to apply to the entire group of patients who experienced early recurrence of the disease [3,4]. However, studies have shown that the POD24 group is heterogeneous: patients with progression in the first 12 months (POD12) and those with relapses and histological transformation within 24 months (POD24 with transformation) are considered to be at ultra-high risk. These patients tend to be resistant to all therapies and have a poor prognosis, with death often occurring within a year of diagnosis. On the other hand, the survival rate for patients with POD24 who do not progress or transform is not significantly different from the overall cohort of FL patients [5]. The search for predictive markers that can help determine the risk of POD12 and POD24 with transformation is a pressing challenge in modern research.

Another issue that illustrates the heterogeneous nature and ambiguity of the general diagnosis of follicular lymphoma (FL) is the subgroup of tumors that do not have a rearrangement of the *BCL2* gene. We know that the 100% diagnostic value of the (q32; q21) t(14;18) translocation has been reconsidered and a subgroup of (14;18)-negative FL has been defined [6]. The pathogenesis of FL 14;18-negative has not been fully understood, and the initiating genetic event has not been identified [7]. FL without *BCL2* gene rearrangement can be divided into two main types: nodal and alternative variants, more common in young patients (pediatric type of FL, primary cutaneous lymphoma from follicular center cells, primary testicular lymphoma) [8]. Nodal FL (14;18)-negative includes the diffuse variant with CD23 expression and a mutation in the *STAT6* gene (CD23+ mut*STAT6*+), as well as FL 3A−3B (14;18)-negative. The CD23+ mut*STAT6*+ FL subgroup has been most studied, and its clinical course is similar to that of classical FL (14;18)-positive [7]. In contrast to indolent CD23+mut*STAT*6+ FL (14;18)-negative, an aggressive course is usually observed in FL 3A–3B (14;18)-negative, which obviously indicates a different biology of these tumors [6]. There are suggestions that FL 3A–3B (14;18)-negative has a closer origin and mechanism of lymphomagenesis to DLBCL (diffuse large B-cell lymphoma) than to classic FL (14;18)-positive [6]. However, there is currently no evidence to support this assumption, mainly because direct comparative studies between FL 3A–3B (14;18)-negative and FL (14;18)-positive are scarce.

A unique feature of FL is spatial (different morphological variants of FL in one node or in two synchronously taken biopsies) and temporal (different picture of the tumor in successive biopsies) heterogeneity. This is a serious obstacle to identifying biological risk factors, potential targets for targeted therapy, and implementing a precision approach to treatment selection [9,10]. Modern predictive scales based on targeted sequencing, although expensive, are characterized by low accuracy. Most of the gene lesions analyzed in these scales are present in a subclonal form in the tumor and differ between different sites of FL [8]. Whereas the prognostic assessment should preferably be based on genetic abnormalities that are present in all tumor cell populations [8,10].

Mutations in the *CREBBP* gene (coding for CREB Binding Protein, a transcriptional coactivator), located at the 16p13.3 locus, occur in approximately 35–75% of patients with FL [8]. Phylogenetic analysis of serial biopsies has identified a *CREBBP* gene mutation as an early event in the evolution of the tumor genome in FL. This mutation appears already in early premalignant cell precursor [10,11]. The high frequency and uniform distribution of the *CREBBP* gene mutations among all tumor subpopulations, both in the initial biopsy and in case of recurrence, suggests that it may be a key driver event alongside t(14;18) [10]. It is known that in FL (14;18)-negative CD23+ mut*STAT*6+, the frequency of the *CREBBP* gene mutations does not differ from that in FL (14;18)-positive. This suggests that at least one early triggering event may be common in these two tumor variants [12]. For FL 3A–3B (14;18)-negative, there is currently no data available on the *CREBBP* gene mutation profile.

Dreval K. et al. compared data from genome-wide sequencing of FL (with and without histological transformation) and DLBCL de novo. They identified two genetically distinct subgroups of FL: DLBCL-like FL and “constrained” FL. These subgroups differed significantly in the frequency of transformation and the time until its development [9]. When comparing the mutation profiles of the two subgroups of FL, striking differences were observed in the pattern of the *CREBBP* gene mutations. In constrained FL, there was an increased number of missense mutations in the lysine acetyltransferase (KAT) domain—70% of cases. In contrast, in DLBCL—like FL and de novo DLBCL, nonsense and frameshift *CREBBP* gene mutations were more prevalent. The authors propose that single missense mutations of *CREBBP* gene limit the mechanisms of transformation in FL and may help to distinguish between groups with a high and low risk of progression [9]. Unfortunately, this study only included cases of FL (14;18)-positive and it did not provide information about the therapy patients received.

Here we present high-throughput sequencing (NGS) data for the *CREBBP* gene in two subgroups of patients: FL 1–3B (14;18)-positive and FL 3A–3B (14;18)-negative. We also report an assessment of the prognostic value of the *CREBBP* gene mutation status for risk of early progression and transformation in FL patients, depending on the treatment option (standard versus high-dose therapy).

## 2. Results

### 2.1. Clinical Data

The main clinical characteristics of the patients are presented in Table 1. In the group of FL 1–3B (14;18)-positive, 31 (50%) patients received treatment according to conventional chemotherapy programs/CCT R-CHOP (cyclophosphamide, doxorubicin, vincristine, rituximab, prednisone) or BR (bendamustine, rituximab) in the first line, another 25 (39%) patients received high-dose chemotherapy with autologous hematopoietic stem cell transplantation (HDCT/auto-HSCT) [13], and 8 (13%) patients underwent unconventional courses. A total of 4 (17%) patients received the CCT (R-CHOP) and 20 (83%) received the Lena-mNHL-BFM-90 HDCT (cyclophosphamide, doxorubicin, vincristine, rituximab, dexamethazone, methotrexate, ifosfamide, etoposide, cytarabine, lenalidomide) in the group FL 3A–3B (14;18)-negative [14].

### 2.2. FISH

Rearrangements of the *BCL6* and *MYC* gene locus were detected more frequently in the FL 3A/3B (14;18)-negative group than in FL 1–3B (14;18)-positive: 9/22 (41%) and 3/22 (14%) cases versus 6/61 (10%) and 2/61 (3%), respectively. Combined rearrangement of the *BCL2, MYC*, and *BCL6* genes was detected in one patient, and combined rearrangements of *MYC* and *BCL6* were detected in another one patient.

### 2.3. Structural Changes in the CREBBP Gene

*CREBBP* gene mutations were detected in 32 (52%) patients with FL 1–3B (14;18)-positive and in 2 (8%) patients with FL 3A−3B (14;18)-negative. We noticed a different pattern of *CREBBP* gene mutations in these groups. In the samples of patients with FL 1–3B (14;18)-positive, missense mutations of the *CREBBP* gene in the KAT domain prevailed and were detected in 20/32 (63%) cases (Figure 1A). *CREBBP* KAT missense mutation was not detected in any case among patients with FL 3A/3B 14;18-negative group (Figure 1B). In 12/32 (37%) patients from the FL 1–3B (14;18)-positive group and in 2/2 (100%) from the FL 3A/3B 14;18-negative group, *CREBBP* gene mutations were represented by nonsense mutations, frameshift mutations, in-frame deletions and insertions, multiple mutations, or any variations outside KAT domain (Appendix A). The majority of *CREBBP* gene mutations were amino acid substitutions at positions p.D1435-S1436 (n = 7), p.S1680 del (n = 6), p.W1502-Y1503 (n = 5), and p.R1446 (n = 4).

Among patients with a missense mutation in the KAT domain, a variant allele frequency (VAF) of more than 50% was detected in six patients indicating possible loss of heterozygosity (LOH). In five out of six cases (83.3%), LOH was detected at the 16p13.13 locus, mapped within 4.7 Mb from the 16p13.3 locus, where the *CREBBP* gene is located. The average LOH value was 63.8%, which roughly corresponded to the average VAF value in this group (62.9%). Thus, five out of six patients with a missense mutation in the KAT domain and a high VAF had chromosomal aberrations affecting the short arm of chromosome 16.

In order to verify the somatic origin of the p.S1680del mutation of the *CREBBP* gene in our sample, the germinal variant of which is associated with the development of Rubinstein-Taubi syndrome, the *CREBBP* gene was sequenced using DNA from blood leukocytes/bone marrow of these patients in remission. The absence of a tumor population in these cells was confirmed by flow cytometry data and a B-cell clonality assay. In five out of six cases the mutations were identified as somatic. In one case, the origin of the mutation could not be determined due to the unavailability of germline material.

### 2.4. Comparison of the Main Characteristics of Patients in the FL 1–3B (14;18)-Positive Group With and Without CREBBP KAT Missense Mutations (KATmiss)

Comparative univariate statistical analysis of the characteristics of follicular lymphoma (FL) 1–3B (14;18)-positive patients with and without *CREBBP* KAT missense mutations showed no statistically significant differences in the examined characteristics between these groups, except for one characteristic—SUVmax (standardized uptake value): in the group of patients with the KAT missense mutation, SUVmax values were significantly lower than in the group without the mutation. It can be assumed that there is a relationship between SUVmax and KAT missense mutation status, but this requires more advanced analysis for clarification. (Table 2).

### 2.5. Survival Analysis

The 5-year EFS rates in FL 3A−3B (14;18)-positive and FL 3A–3B (14;18)-negative groups were 46% (95% confidence interval [CI] 33–65) and 78% (95% CI 61–100), respectively (Figure 2a,b). The median follow-up for patients with FL 1–3B (14;18)-positive and for patients with FL 3A–3B (14;18)-negative was 50 month and 61 months, respectively. The average time to relapse in the FL 3A–3B (14;18)-negative group was 11 months versus 25 months in the FL 1–3B (14;18)-positive group. After HDCT, relapse developed in 10% (2/20) of patients and in 100% (4/4) of patients after R-CHOP chemotherapy in the FL 3A–3B (14;18)-negative group. In the group of FL 1–3B (14;18)-positive failures, relapses/progressions were observed in 4 (16%) out of 25 patients after HDCT/auto-HSCT, in 14 (44%) out of 32 patients after CCT, and in 4 (57%) out of 7 patients after non-conventional courses.

Next, we compared OS in the subgroups of patients with FL 1–3B (14;18)-positive with and without KATmiss–, depending on the treatment option (HDCT/auto-HSCT vs. CCT). In patients with FL 1–3B (14;18)-positive with KATmiss, the 5-year OS was 100% for both HDCT/auto-HSCT and the CCT (median follow-up of 30 and 26 months, respectively) (Figure 3a,c). In the CCT group, 5-year EFS was lower in the FL subgroup FL 1–3B (14;18)-positive without KATmiss compared with FL 1–3B (14;18)-positive with KATmiss+: 33% (95% CI, 15–71) vs. 49% (95% CI, 27–88) (median follow-up of 37 months) (Figure 3b). In the HDCT/auto-HSCT group, 5-year EFS was higher in the group without the KAT mutation (median follow-up of 43 months) (Figure 3d). The comparatively higher EFS under HDCT/auto-HSCT in patients without the KAT domain missense mutation is most likely attributable to sample bias due to the small number of events in the compared patient subgroups. For the FL 1–3B (14;18)-positive without KATmiss, 5-year EFS was significantly higher in the cohort of patients who received HDCT/auto-HSCT than in the CCT group: 86% (95% CI, 71–100) and 33% (95% CI, 15–71), respectively (*p* = 0.05) (Figure 3e). There was a trend to increased OS (5-year OS 94% (95% CI, 83–100) vs. 77% (95% CI, 52–100) (Figure 3a,c). For FL 1–3B (14;18)-positive with KATmiss patients, the rates of 5-year OS and EFS in both treatment groups were comparable: 5-year OS 100% vs. 100% (Figure 3a,c) and 5-year EFS 49% (95% CI, 27–88) vs. 60% (95% CI, 29–100), *p* = 0.47) (Figure 3b,d). To assess the presumed protective effect of the KATmiss mutation against adverse outcomes, we calculated the hazard ratio (HR) for the cases with the highest number of events (EFS under conventional programs, Figure 3b). The calculated HR (HR = 0.54, *p* = 0.22) showed a trend toward benefit (since HR < 1), although it did not reach statistical significance, which is consistent with the assumption that the KATmiss mutations may have a protective effect.

### 2.6. Causes of Death

Deaths occurred in six patients and were associated with lymphoma. All but one death was in FL 1–3B (14;18)-positive without KATmiss.

### 2.7. Frequency and Time to Transformation, Frequency of Progression

The frequency of POD12 in the group of FL 1–3B (14;18)-positive with KATmiss was 7%, while in the group of FL 1–3B (14;18)-positive without KATmiss, it was 39% (Figure 2b). The frequency of POD12 was lower after HDCT/auto-HSCT compared with CCT—7% versus 38%, respectively. In 13 patients with FL 1–3B (14;18)-positive repeated biopsies were performed in the tumor recurrence. Of these, histological transformation was confirmed in six (46%) cases, including five cases with an early transformation (POD24-transf.). The median time to transformation was 10 months. Transformation was not detected in any of the seven repeated biopsies among patients with FL 1–3B (14;18)-positive with KATmiss (Figure 4). All cases of transformation occurred in the group of CCT or non-conventional chemotherapy. In contrast, there were no cases of transformation during HDCT/auto-HSCT in groups with or without KATmiss.

## 3. Discussion

We found that assessing the mutation status of the *CREBBP* gene can help differentiate between aggressive ultra-high-risk FL and indolent FL with a favorable prognosis. In the FL 1–3B (14;18)-positive group with the single KAT missense mutation, there was an extremely low incidence of transformation, progression and early death from lymphoma regardless of the treatment option. Furthermore, the detection of a missense mutation in the KAT domain can be considered as a predictive marker of response to CCT. In patients with FL 1–3B (14;18)-positive without the KAT missense mutation poor effectiveness of conventional regimens was observed. The use of HDCT/auto-HSCT in this group of patients significantly improved EFS by reducing the frequency of transformation and POD12. Moreover, the difference in OS between HDCT/auto-HSCT and CCT in patients with FL 1–3B (14;18)-positive without KATmiss was nearly significant (*p* = 0.09). Reducing the frequency of transformation and POD12 is extremely important, since the majority of lymphoma-related deaths occur in this cohort of patients. In a large FL study, GALLIUM C.Casulo and colleagues showed that the survival rates were the worst in the POD24 group with histological transformation (2-year OS 45%) and in the POD12 group regardless of the presence of transformation (2-year OS 43%). At the same time, OS and PFS were comparable in patients with POD24 without transformation and in the general cohort (2-year OS 80%) [8].

It should be noted that the FL 1–3B (14;18)-positive with and without KATmiss did not differ in any of the other clinical or laboratory parameters, i.e., standard prognostic scores probably cannot distinguish these groups.

Currently, a lot of information has been accumulated about the pathogenesis of FL. In particular, it was found that a single *BCL2* rearrangement/t(14;18) is not enough to cause a lymphoma. Additional driver alterations are necessary for the appearance of a tumor and 90% of patients with FL have certain mutations in genes regulating transcription and chromatin remodeling such as *CREBBP, EP300, KMT2D* and *EZH2* [6,7]. Missense mutations of the *CREBBP* gene appear at the early stages of the evolution of the tumor genome [10,15]. A number of authors hypothesize that the *CREBBP* gene mutation is an earlier event than t(14;18) [15,16]. This hypothesis is supported by cases of FL 1–3B (14;18)-positive in patients with a germinal mutation of the *CREBBP* gene (Rubinstein-Taubi syndrome) [16]. However, no other data supporting this hypothesis has been described in the literature. We have demonstrated the somatic status of *CREBBP* gene mutations in all but one patient with VAF of more than 50%. Since the missense mutation of the *CREBBP* gene suspected to appear in the early tumor precursors, it presents in all malignant cells and mutation analysis in this gene can be performed at any stage of disease development (initial or relapse) and in any tumor site. Therefore, detection of *CREBBP* gene mutations provides an opportunity to stratify FL patients in routine clinical practice.

In our study 63% of single missense mutations in the KAT domain of *CREBBP* gene were detected in FL 1–3B (14;18)-positive patients.

The mechanisms underlying the effects of single missense mutations in the KAT domain of *CREBBP* gene are not fully understood. Most research until recently has been focused on the effect of *CREBBP* gene knockout rather than missense mutations. It is known that in the *CREBBP* gene with its *EP300* paralog, encode enzymes that act as global transcription coactivators, interacting with more than 400 transcription factors by modifying (acetylating) lysine. In cases of *CREBBP* disorders, the activity of transcription factor enhancers is still preserved, ensured by compensatory activation of *EP300*. *CREBBP* acetylates lysine (K) at the 27th and 18th positions of histone H3 (H3K27 and H3K18), which leads to chromatin-dependent gene derepression. Thus, *CREBBP* modulates the expression of many genes that are involved in the exit of the centrocyte from the germinal center, including the signaling pathways of the B-cell receptor, CD40 receptor, plasma cell regulator IRF4, as well as the processing and presentation of antigen through the main histocompatibility complex class II (MCH II) [17,18].

Currently it has become obvious that the effect of missense mutations in the KAT domain is fundamentally different from the effect of a loss of function mutations and is consistent with the differences in the clinical course of FL [19]. Nonsense and frameshift mutations cause a decrease in the enzyme and initiate the development of lymphoma which phenotypically mimics DLBCL or aggressive FL with overexpression of *MYC* and active aberrant somatic hypermutation (aSHM) [9,20]. In contrast, the *CREBBP* gene with missense mutation expressed into functional protein with altered KAT domain. The most important feature of altered *CREBBP* protein is the ability to prevent compensatory hyperactivation of its *EP300* paralog. Such enzymes are called zombie enzymes, they are catalytically inactive [20]. As a result of the production of altered CREBBP, centrocyte enhancers remain inactive despite CD40 simulation, and the transition from centrocyte to centroblast is disrupted [9,10,20,21]. Unlike *CREBBP* gene inactivating mutations, *MYC* hyperexpression was never observed in patients with KAT missense mutations, and the activity of aSHM was less pronounced than in patients without KAT missense mutations, which may explain why the transformation is rarely observed in these patients [9]. On the other hand, in FL with missense KAT mutations, a 10-fold decrease in MHC class II expression was observed compared with DBCCL/FL with *CREBBP* inactivation. Thus, tumors provide an opportunity to evade the immune response through reduced antigen presentation [22]. In mouse models, it has been shown that chemotherapy regimens such as CHOP are able to restore the loss of MHC-I/II loss in tumor cells. One can speculate that this explains the high effectiveness of standard chemotherapy for FL with the KAT mutation [23].

For the first time we analyzed the mutation profile of the *CREBBP* gene in patients with FL 3A−3B (14;18)-negative. It is well known that FL 3A−3B (14;18)-negative differs from the classic FL [6,7]. Since no practical recommendations for the management of FL 3A−3B (14;18)-negative are formulated in any guidelines or classification, these patients are examined and treated according to the general standards for classic FL [12]. The main parameters from the generally accepted FLIPI and FLIPI2 scales, such as stage, hemoglobin level and bone marrow involvement may not be valid for determining the prognosis for patients with FL 3A−3B (14;18)-negative. In all patients with FL 3A−3B (14;18)-negative, the standard R-CHOP regimen proved ineffective; on the contrary, therapy according to the HDCT NHL-BFM scheme allowed achieving remissions in most patients. Also, according to our data, all relapses were early in patients with FL 3A−3B (14;18)-negative, in contrast to classic FL, for which late relapses are typical. This behavior of the tumor is more similar to DLBCL than to indolent FL. In our work, for the first time, a fundamental difference was demonstrated in the clinical course of FL (14;18)-positive and FL 3A−3B (14;18)-negative. One can speculate that in the absence of t(14;18) and the KAT missense mutation, FL 3A−3B (14;18)-negative may originate from distinct precursor cells lacking t(14;18) and the *CREBBP* gene mutation. Our hypothesis needs to be tested in a larger sample of patients with (14;18)-FL. However, we note that Sun et al. obtained similar results in the larger FL3B subgroup, although the authors did not analyze the relationship between *BCL2* rearrangement and *CREBBP* mutations [24].

Based on the data obtained, we propose to divide FL into 4 types: Type 1 FL (14;18)-positive with missense KAT mutation; Type 2: FL (14;18)-positive without missense KAT mutation; Type 3: FL (14;18)-negative with missense KAT mutation (we have not seen any patients from type 3 in our study yet); Type 4: FL (14;18)-negative without missense mutation KAT (Figure 5). The FL type 1 (and possibly 3) expected to have an indolent course and a good prognosis, whereas types 2 and 4 are high-risk FL subtypes that require a special approach to treatment. We believe that type 2 FL requires the use of HDCT/auto-HSCT, early use of CAR-T therapy or bispecific antibody therapy, while FL type 4 should be treated by short intensive courses (for example, NHL-BFM-like programs). The algorithm for approximating FL subgroups that we proposed is hypothetical, since it is based on a limited data set and can only be considered as a model for further generation of hypotheses.

Our study has a number of limitations. Firstly, our cohort is relatively small and selected and all the results obtained need to be tested in a larger cohort of patients with prospective validation. Secondly, there is an obvious bias towards patients with progression and relapses in the FL cohort under study since we selected such patients in order to test the negative prognostic value of *CREBBP* gene mutations. In the future, it is necessary to expand the subgroup of patients with FL 1–3B (14;18)-positive with long-term remissions on conventional programs and check whether the prognostic value of determining KAT mutations remains not only in the mixed group of standard therapy but also in cohorts of patients who received a certain treatment (R-CHOP or BR, etc.). In order to exclude the effect of selection bias, it is necessary to test the obtained data on balanced subgroups of patients who received HDCT/auto-HSCT and CCT.

The frequency of transformation in our work may be underestimated because not all patients with relapse/progression underwent biopsy. We also did not have the opportunity to conduct additional studies such as determination of IRF4 gene rearrangements in patients with FL 3A–3B (14;18)-negative with MUM-1 expression, determination of *STAT6/TNFRSF14* gene mutations in FL (14;18)-negative with CD23 expression, evaluation of MYC expression and aSHM in FL samples with *CREBBP* gene deletion.

## 4. Materials and Methods

### 4.1. Patients

Archival tumor samples obtained before treatment initiation from 86 FL patients who attended the National Medical Research Center for Hematology (Moscow, Russia) from 2013 to 2024 were included in this study. Freshly frozen or formalin-fixed paraffin-embedded (FFPE) tumor samples from 24 FL 3A–3B (14;18)-negative and 62 FL 1–3B (14;18)-positive were analyzed. All patients were diagnosed according to the classification criteria of the World Health Organization (2017) [1]. Morphology, growth pattern, cytology, and immunohistochemical staining for CD20, CD79a, BCL6, BCL2, CD10, CD3, MUM-1, C-MYC, and CD23 were evaluated in FFPE tissue sections.

### 4.2. DNA Isolation, Clonality Assessment, and BCL2::IGH Breakpoints Evaluation

DNAs from the FFPE samples and freshly frozen samples were extracted essentially as described in [25]. B-cell clonality in the *IGH* (FR1, FR2, and FR3) and *IGK* (VK-Jk and VK/intron-Kde) genes was analyzed using PCR with BIOMED-2 multiplex primers and subsequent fragment analysis by capillary electrophoresis on a Nanophore-05 genetic analyzer (Institute of Analytical Instrumentation, Saint Petersburg, Russia) [26]. The analysis of the *BCL2::IGH* breakpoints was performed using real-time PCR.

### 4.3. Fluorescence In Situ Hybridization (FISH)

In situ interphase fluorescent hybridization analysis to detect BCL2, BCL6, and MYC translocations was performed using LSI BCL2, BCL6, and MYC Dual-Color Break-Apart probes (Vysis-Abbott Molecular, Wiesbaden, Germany).

### 4.4. NGS

In all 86 patients, targeted sequencing of exons 22–30 of the *CREBBP* gene was performed using MiSeq System (Illumina, San Diego, CA, USA), with the MiSeq Reagents Kit v2—300 cycles (Illumina, San Diego, CA, USA). Before sequencing, a B-cell clonality study was performed to assess the presence of the tumor population in all samples. Data filtering, deletion of service sequences, mapping reads to the reference genome, search for variants, calculation of VAF (variant allele frequencies) and variant annotation were carried out using Trimmomatic, BWA, SAMtools, Vardict, and Annovar utilities and annotated according to the dbSNP, ClinVar, and COSMIC databases [27,28,29,30,31,32,33,34]. To analyze the obtained results, we followed the recommended minimum threshold values of the coverage depth for the VAF values corresponding to different detection limits (LOD). For LOD 10%, we used 175 reads, for LOD 5%, 562 reads, and for LOD 3%, 1650 reads. With these threshold values, a 99.9% chance of obtaining a true positive result was achieved [35]. Variants that did not meet these criteria, as well as those with a VAF below 3%, were excluded from the analysis. The oncogenic potential of *CREBBP* gene variants was assessed in accordance with the ClinGen-CGC-VIC guidelines [35,36].

### 4.5. LOH

The loss of heterozygosity on the short arm of chromosome 16 was investigated using the STR-PCR technique. A set of primers for microsatellite loci 16p13.13 (CA)n and (GT)m, located between the CIITA and SOCS1 genes, was used [37].

### 4.6. Statistical Analysis

Continuous variables were assessed for normality using the Shapiro–Wilk test and are presented as medians with ranges or interquartile ranges or as means with standard deviations (SD), as appropriate. The comparison of continuous variables was performed using the Mann–Whitney U test. The categorical variables were summarized as absolute numbers and percentages and compared between groups using the chi-squared test, Fisher’s exact test, or the Cochran–Mantel–Haenszel test, as appropriate. Overall survival (OS) and event-free survival (EFS) were estimated using the Kaplan–Meier method (with right-censored data). The events were defined as death, relapse, disease progression, failure to achieve remission, treatment resistance, and therapy modification. The survival times were measured from the initiation of treatment. The log-rank test was used to compare survival curves between the patient groups. The proportional hazards assumption was tested using Schoenfeld’s test, which confirmed the comparability of the survival curves. Hazard ratio (HR) was calculated in cases where the number of events was sufficient. The median follow-up time for survival was estimated using the reverse Kaplan–Meier method. To control the false discovery rate, the p-values were adjusted for multiple comparisons. Statistical significance was set to *p* ≤ 0.05. Data analysis was performed using SPSS 25.0. The analysis was based on data collected as of 12 December 2024.

## 5. Conclusions

The detection of single missense mutations in the KAT domain of *CREBBP* gene can be used as a predictive marker for patients with classic FL. We argue that patients with FL (14;18)-positive without a single missense mutation in the KAT domain can be considered as candidates for intensive treatment options in the first line (HDCT/auto-HSCT, CAR-T therapy, bispecific antibodies, etc.). FL 3A–3B (14;18)-negative might be considered a distinct disease, different from classic FL, both in clinical course and in molecular genetic characteristics, including the mutation profile of the *CREBBP* gene.

## Figures and Tables

**Figure 1 ijms-26-06913-f001:**
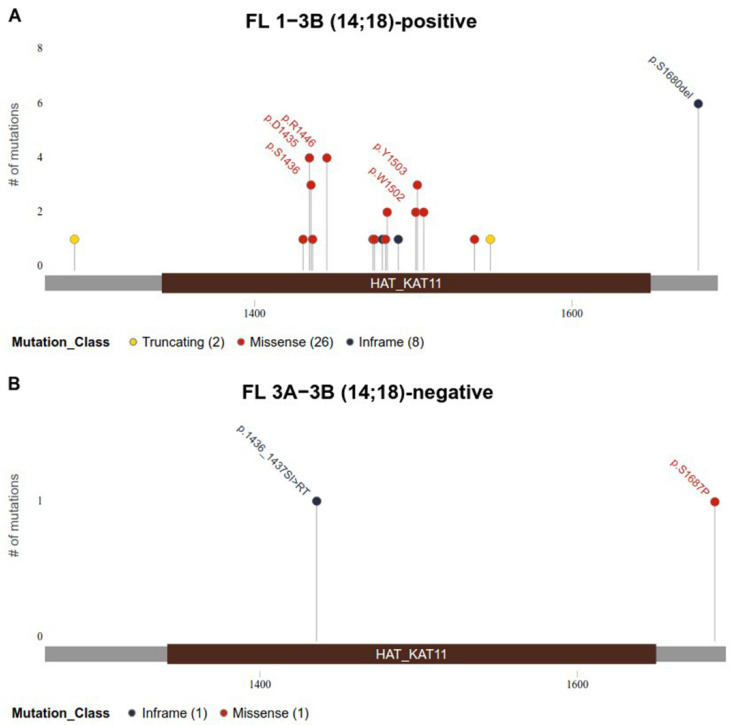
Identified mutations in the KAT domain (1342–1649 bp) of *CREBBP* gene (reference transcript NM_004380.3). (**A**). Mutations identified in FL 1–3B (14;18)-positive group. (**B**). Mutations identified in FL 3A−3B (14;18)-negative group. Mutations are colored based on their type. Each mutation is annotated with amino acid substitution.

**Figure 2 ijms-26-06913-f002:**
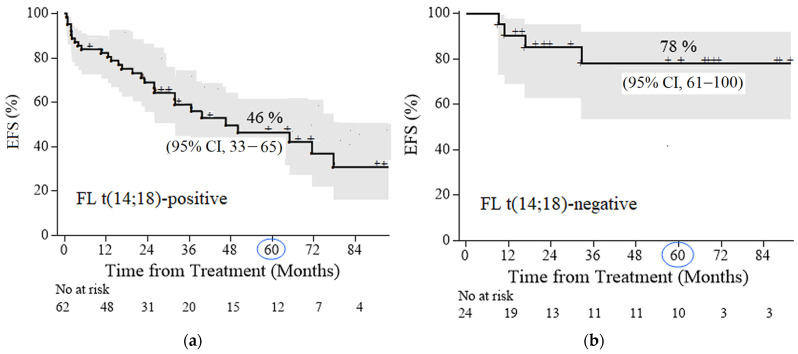
Event-free survival (EFS) in follicular lymphoma (FL) patients: (**a**) FL 1–3B (14;18)-positive, with a median follow-up of 50 months (**b**) FL 3A−3B (14;18)-negative, with a median follow-up of 61 months.

**Figure 3 ijms-26-06913-f003:**
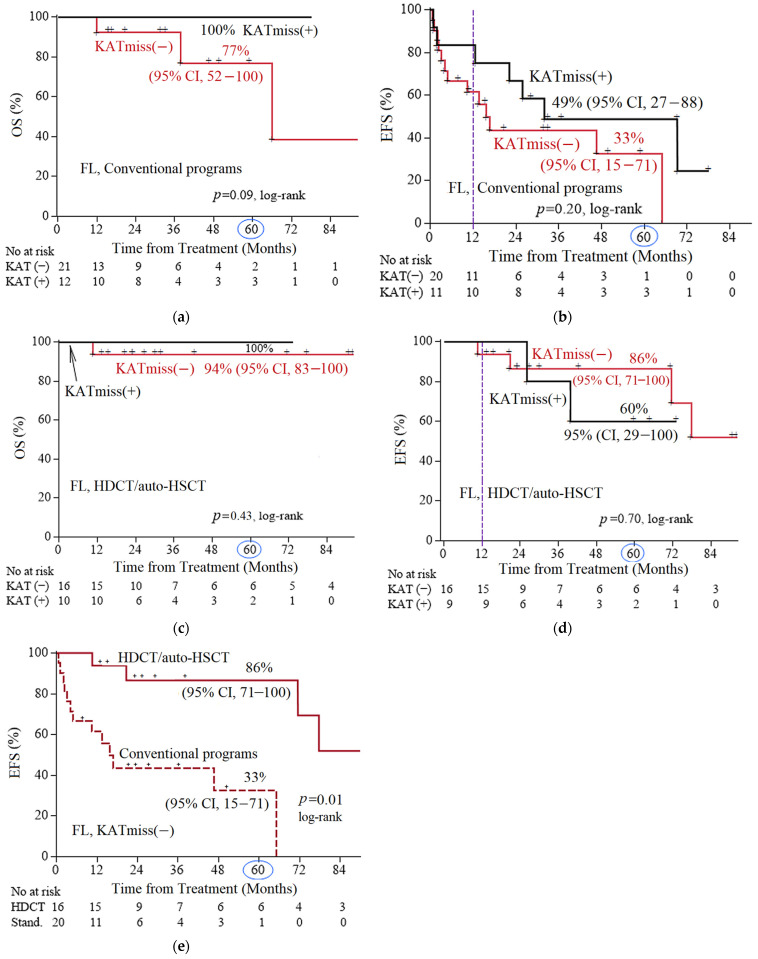
Comparison of 5-year survival rates among different subgroups of follicular lymphoma (FL) 1–3B (14;18)-positive patients. (**a**) Comparison of overall survival (OS) according to the status (presence [+] or absence [−]) of CREBBP KAT missense mutation (KATmiss) in patients receiving conventional chemotherapy programs; (**b**) comparison of event-free survival (EFS) according to the KATmiss status in patients receiving conventional programs; (**c**) comparison of OS according to the KAT status in patients receiving high-dose chemotherapy with autologous hematopoietic stem cell transplantation (HDCT/auto-HSCT); (**d**) comparison of EFS according to the KATmiss status in patients receiving HDCT/auto-HSCT; (**e**) comparison of EFS according to the therapy type: conventional chemotherapy versus HDCT/auto-HSCT (both to the case of KATmiss(−)). The point on the time axis corresponding to 5 years (60 months) was marked with a circle; censored data was indicated by plus signs (+) on the survival curves.

**Figure 4 ijms-26-06913-f004:**
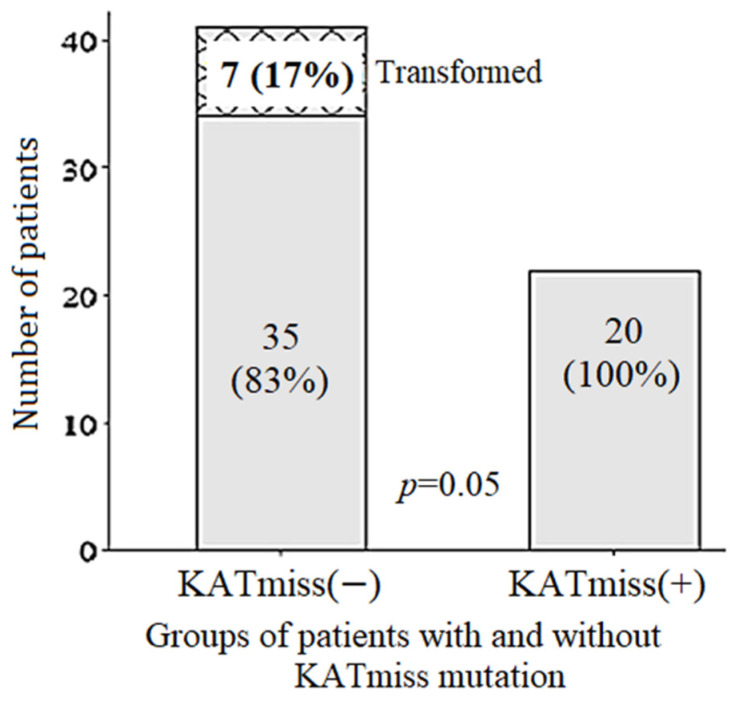
Transformation of follicular lymphoma in 1–3B (14;18)-positive patients with (n = 20) versus without (n = 42) *CREBBP* KAT missense mutation (KATmiss).

**Figure 5 ijms-26-06913-f005:**
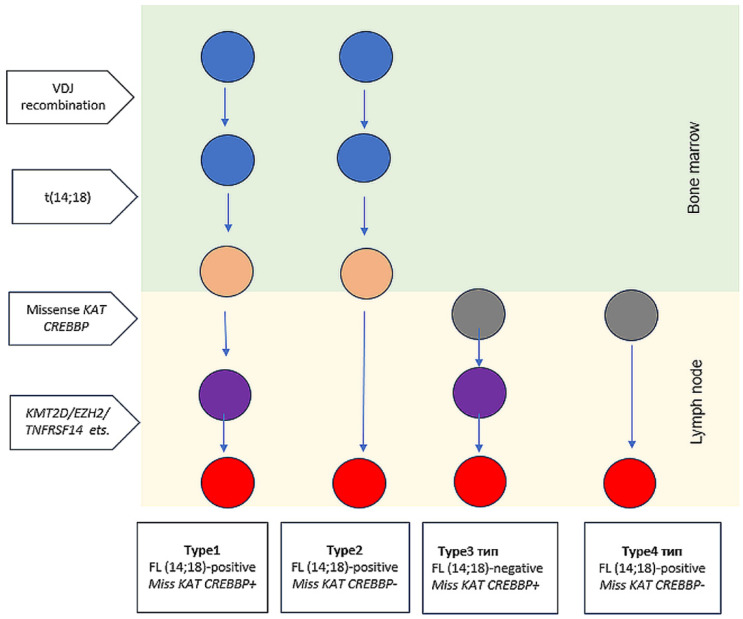
Proposed FL types: Type 1 FL (14;18)-positive with missense KAT mutation; Type 2: FL (14;18)-positive without missense KAT mutation; Type 3: FL (14;18)-negative with missense KAT mutation; Type 4: FL (14;18)-negative without missense mutation KAT. The colors of the circles indicate the cell type: blue—B-cell progenitor; gray—lymph node non-malignant lymphocyte, orange—non-malignant B-cell clone; purple—premalignant tumor precursor; red—early malignant clone. Non-malignant B-cell clone, premalignant tumor precursor and early malignant clone refer to early lymphoma precursor.

**Table 1 ijms-26-06913-t001:** Comparison of the FL 1–2/3A (14;18)-positive and FL 3A–3B (14;18)-negative groups.

Characteristics	FL 1–3B (14;18)-Positive (n = 62)	FL 3A–3B (14;18)-Negative (n = 24)	*p-*Value
Male/Female	31/31	14/10	0.63
Age, median (range). years	42 (24–77)	47 (34–71)	0.21
IV stage	2/62 (84%)	6/23 (26%)	0.001
Local involvement of the lymph. nodes of the head and neck (tonsils +/−)	0/62 (0%)	12/24 (50%)	0.006
ECOG:			0.94
0–1	38/62 (61%)	9/23 (39%)	
2	16/62 (25%)	7/23 (30%)	
3	7/62 (11%)	3/23 (13%)	
4	2/62 (3%)	4/23 (18%)	
Bulky (>6 cm)	38/61 (62%)	11/24 (45%)	0.22
LDG, higher/total	32/58 (55%)	11/24 (52%)	0.47
Hemoglobin ˂120 g/L	39/62 (63%)	8/24 (33%)	0.03
Bone marrow involvement	51/62 (82%)	2/24 (9%)	0.001
FLIPI:			0.01
low	6/59 (10%)	12/23 (52%)	
intermediate	18/59 (31%)	3/23 (13%)	
high risk	35/59 (59%)	8/23 (35%)	
SUVmax, median (range) [interquartile range]	10.8 (3.9–49) [8.8–13.6]	18.0 (6.8–49) [10.5–23.9]	0.01
SUVmax, mean (SD)	11.7 (5.0)	18.3 (5.0)	0.01
SUVmax > 14	12/47 (25%)	11/18 (61%)	0.01
SUVmax > 20	7/47 (15%)	8/18 (44%)	0.02

**Table 2 ijms-26-06913-t002:** Comparison of the characteristics of patients in the FL 1–3B (14;18)-positive group with and without CREBBP KAT missense mutations.

Characteristics	FL 1–3B (14;18)-Positive With KATmiss (n = 20)	FL 1–3B (14;18)-Positive Without Katmiss (n = 42)	*p-*Value
Male/Female	9/11	27/15	0.18
Age, median (range). years	47 (32–77)	45 (24–71)	0.31
IV stage	17/20 (85%)	33/42 (79%)	0.80
ECOG:			0.98
0–1	14/19 (74%)	23/42 (55%)	
2	4/19 (21%)	11/42 (26%)	
3	0/19 (0%)	7/42 (17%)	
4	1/19 (5%)	1/42 (2%)	
Bulky (>6 cm)	13/20 (65%)	27/40 (68%)	0.85
LDG. higher/total	13/18 (72%)	21/37 (57%)	0.42
Hemoglobin ˂120 g/L	8/20 (40%)	11/38 (29%)	0.58
Grade:			0.96
1–2	15/20 (75%)	29/42 (69%)	
3A	2/20 (10%)	6/42 (14%)	
mix (1–2 + 3A, 1–2 + 3A)	3/20 (15%)	7/42 (17%)	
Bone marrow involvement	17/20 (85%)	35/41 (85%)	1.00
FLIPI:			0.99
low	3/18 (17%)	4/40 (10%)	
intermediate	5/18 (28%)	13/40 (33%)	
high-risk	10/18 (56%)	23/40 (57%)	
SUVmax, median (range), [interquartile range]	8.6 (4.4–20.1), [5.14–13.18]	11.7 (3.9–22.3), [10.1–18.0]	0.05
SUVmax, mean (SD)	9.7 (4.6)	12.6 (5.0)	0.05
SUVmax > 14	2/14 (25%)	8/31 (29%)	0.71
SUVmax > 20	1/14 (0%)	4/31 (16%)	1.00

## Data Availability

Data available from authors upon reasonable request.

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
