# Peer review of "Missense Mutations in the KAT Domain of CREBBP Gene in Patients with Follicular Lymphoma: Implications for Differential Diagnosis and Prognosis"

_ijms, 2025, doi:10.3390/ijms26146913_

Round 1
Reviewer 1 Report
Comments and Suggestions for Authors
This study investigates the prognostic significance of CREBBP KAT domain missense mutations in follicular lymphoma (FL). The authors show that patients with FL (14;18)-positive harboring a single KAT missense mutation have significantly better outcomes and lower transformation risk, regardless of treatment. In contrast, patients without this mutation benefit more from intensive therapies like HDCT/auto-HSCT. The absence of KAT mutations in FL 3A–3B (14;18)-negative suggests a distinct biological subtype with aggressive behavior. Here are some comments which need to be addressed.
- The manuscript suggests that CREBBP KAT missense mutations could guide frontline treatment decisions (e.g., CCT vs. HDCT/auto-HSCT). However, the cohort is small and highly selected, and prospective validation is lacking. The author can added some discussion about their limitations.
- The comparison between CREBBP mutation patterns in FL (14;18)-positive vs. negative subgroups is intriguing, but the (14;18)-negative group includes only 24 patients, and just 3 had CREBBP mutations. The author should carefully make the conclusions about their distinct biology.
- Some survival plots (e.g., Figure 3 panels) lack clear labeling, confidence intervals, and proper legends. The coloring is inconsistent, and the axes are not always labeled with units or definitions (e.g., what constitutes EFS).
- The improved survival with HDCT/auto-HSCT in patients without KAT mutations may reflect selection bias rather than mutation-driven treatment resistance. The author should discuss the possibility.
- There are multiple typographical and grammatical errors (e.g., “deesiase” instead of “disease”, inconsistent use of spaces and dashes). The text would benefit from careful English editing to improve clarity and professionalism.
- The proposed 4-type classification system based on KAT mutation and translocation status is conceptually interesting, but overreaches based on the limited dataset. Authors should clearly discuss as a hypothesis-generating model.
Author Response
Response to Reviewer 1 Comments
Dear Reviewer,
Firstly, we would like to express our gratitude for your thorough review of our manuscript. We are sure that your comments will allow us to significantly enhance the presentation of our material and make our manuscript more suitable for publication. Please find below our answers to your suggestions and criticism (typed in red).
Comment 1: The manuscript suggests that CREBBP KAT missense mutations could guide frontline treatment decisions (e.g., CCT vs. HDCT/auto-HSCT). However, the cohort is small and highly selected, and prospective validation is lacking. The author can added some discussion about their limitations.
Response 1: Thank you for pointing this out. Indeed we failed to discuss the limitations of our study. Appropriate text is added at lines 373-375.
Comment 2: The comparison between CREBBP mutation patterns in FL (14;18)-positive vs. negative subgroups is intriguing, but the (14;18)-negative group includes only 24 patients, and just 3 had CREBBP mutations. The author should carefully make the conclusions about their distinct biology.
Response 2: Yor are absolutely right. Our hypothesis needs to be tested in a larger sample of patients with 14;18- FL. However, we note that Sun et al. obtained similar results in the larger FL3B subgroup, although the authors did not analyze the relationship between BCL2 rearrangement and CREBBP mutations. We now discuss this issue at lines 349-352.
Comments 3: Some survival plots (e.g., Figure 3 panels) lack clear labeling, confidence intervals, and proper legends. The coloring is inconsistent, and the axes are not always labeled with units or definitions (e.g., what constitutes EFS).
Response 3: Thank you for bringing this to our attention. The appropriate descriptors have now been added to the figures 2, 3 and the figure legends at lines 231-238 and 255-256.
Comments 4: The improved survival with HDCT/auto-HSCT in patients without KAT mutations may reflect selection bias rather than mutation-driven treatment resistance. The author should discuss the possibility.
Response 4: We agree with this comment. We have added discussions of this issue at lines 381-383 and 212-215; Given the potentially possible selection bias (due to small quantity of data), for Hazard Ratio calculation (which we present now) we selected comparison of survival curves with the maximal number of events – lines 222-227.
Comments 5: There are multiple typographical and grammatical errors (e.g., “deesiase” instead of “disease”, inconsistent use of spaces and dashes). The text would benefit from careful English editing to improve clarity and professionalism.
Response 5: Apologies for any errors. We have thoroughly checked the entire text for potential misspellings and typos.
Comments 6: The proposed 4-type classification system based on KAT mutation and translocation status is conceptually interesting, but overreaches based on the limited dataset. Authors should clearly discuss as a hypothesis-generating model.
Response 6: The algorithm for approximating FL subgroups that we proposed is hypothetical since it is based on a limited data set and can only be considered as a model for further generation of hypotheses. We have added this clarification at lines 362-364.
Reviewer 2 Report
Comments and Suggestions for Authors
Overall evaluation
Overall, this manuscript provides a thorough analysis of missense mutations in the KAT domain of the CREBBP gene among patients with follicular lymphoma (FL), along with their implications for prognosis and treatment strategies. The study design is well-structured, and the data are presented in sufficient detail. The findings that KAT domain missense mutations may serve as prognostic markers and that patients harboring these mutations tend to have a better prognosis hold clinical significance, offering novel insights into personalized FL treatment. However, certain limitations exist, such as a relatively small sample size, insufficient depth in some discussions, and methodological constraints in statistical analyses and literature citations, which warrant further refinement.
Specific problems and suggestions for improvement
Page 3, Table 1. The manuscript reports a sample size of 86 cases, which is modest and could compromise the robustness and reliability of the results. Specifically, after stratification, there are 62 cases in the FL 1-3B (14;18)-positive group and 24 cases in the FL 3A-3B (14;18)-negative group. The limited sample size in the latter group might introduce bias into the results, necessitating cautious interpretation.
Page 12, "4.6. Statistical Analysis" section. While the manuscript elaborates on the statistical methods employed, it lacks a comprehensive explanation of the rationale and applicability of certain complex methods, such as the log-rank test used in survival analysis. Providing additional justification would enhance the transparency and validity of the statistical approach.
Pages 9-10, "3. Discussion" section. Although the discussion section partially elucidates the research findings, the exploration of critical mechanisms, such as the relationship between KAT domain missense mutations and favorable prognosis, remains superficial. A more in-depth analysis would strengthen the interpretative power of the results.
Pages 5-7, 9. Figures like Figure 3 involve comparisons across multiple subgroups and survival rates, requiring more detailed annotations for clarity. Furthermore, the current number of figures may be insufficient to fully capture all aspects of the study's findings, suggesting the inclusion of additional visual aids.
Pages 14-15, "References" section. While key studies are referenced, some cited works appear outdated and fail to reflect recent advancements in the field. Expanding the scope of citations to include more contemporary and comprehensive sources would improve the manuscript's relevance and comprehensiveness.

The English could be improved to more clearly express the research.
Author Response
Response to Reviewer 2 Comments
We would like to express our deepest gratitude for your attention to this manuscript. We hope your helpful comments and changes we do accordingly will improve the presentation of our work and push it closer to publication. Please, find responses below and appropriate changes in manuscript marked in red.
Comments 1: Page 3, Table 1. The manuscript reports a sample size of 86 cases, which is modest and could compromise the robustness and reliability of the results. Specifically, after stratification, there are 62 cases in the FL 1-3B (14;18)-positive group and 24 cases in the FL 3A-3B (14;18)-negative group. The limited sample size in the latter group might introduce bias into the results, necessitating cautious interpretation.
Response 1: Thank you for pointing this out. We have added information about the limitations of our study at lines 373-375, lines 381-383, and lines 212-215. Given the potentially possible selection bias (due to small quantity of data), for Hazard Ratio calculation (which we present now) we selected comparison of survival curves with the maximal number of events – line 222-227.
Comments 2: Page 12, "4.6. Statistical Analysis" section. While the manuscript elaborates on the statistical methods employed, it lacks a comprehensive explanation of the rationale and applicability of certain complex methods, such as the log-rank test used in survival analysis. Providing additional justification would enhance the transparency and validity of the statistical approach.
Response 2: We agree that statistical methods should be described in more detail. Additional explanations have been included: lines 438-445 and line 448.
Comments 3: Pages 9-10, "3. Discussion" section. Although the discussion section partially elucidates the research findings, the exploration of critical mechanisms, such as the relationship between KAT domain missense mutations and favorable prognosis, remains superficial. A more in-depth analysis would strengthen the interpretative power of the results.
Response 3: Thank you for bringing this to our attention. We have added additional information regarding the mechanisms of malignant transformation in cases with KAT missense mutations and provided possible explanations for the high effectiveness of standard treatment programs in this patient subgroup: lines 298-332.
Comments 4: Pages 5-7, 9. Figures like Figure 3 involve comparisons across multiple subgroups and survival rates, requiring more detailed annotations for clarity. Furthermore, the current number of figures may be insufficient to fully capture all aspects of the study's findings, suggesting the inclusion of additional visual aids.
Response 4: 1) Some additional explanations have now been added to the text discussing the data presented in Fig. 3. Please see lines 212-215, lines 222-227, lines 438-445, and line 448: 2 In the current study, given the limited dataset, we make no claim to address all aspects. A comprehensive analysis would require further partitioning of the already limited data, which would produce statistically fragile results. However, we understand the issue and plan to continue this research using a more extensive dataset.
Comments 5: Pages 14-15, "References" section. While key studies are referenced, some cited works appear outdated and fail to reflect recent advancements in the field. Expanding the scope of citations to include more contemporary and comprehensive sources would improve the manuscript's relevance and comprehensiveness
Response 5: We agree with this comment and have added more up-to-date and comprehensive sources (references 18, 21-24).
Reviewer 3 Report
Comments and Suggestions for Authors
The authors studied the potential diagnostic and prognostic values of the variants in the KAT domain of CREBBP in patients with FL. They conclude that a single missense mutation in the CREBBP gene may be associated with a favorable prognosis. Although the current data may not be sufficient to support this conclusion due to the low sample size, it could potentially contribute to further investigation of the disease. However, several issues must be addressed for the manuscript to be considered acceptable for publication.
Major:
- In lines 144-150, the authors state, “In the samples of patients with FL 1−3B (14;18)-positive, missense mutations of the CREBBP gene in the KAT domain prevailed and were detected in 20/32 (63%) cases. CREBBP KAT missense mutation wasn’t detected in any case among patients with FL 3A/3B 14;18- group (Figure 1). In 12/32 (37%) patients from the FL 1−3B (14;18)-positive group and in 3/3 (100%) from the FL 3A/3B (14;18)- group, CREBBP gene mutations were represented by nonsense mutations, frameshift mutations, or multiple mutations (Figure 1).” However, Figure 1 only lists the variants identified in the study, making it impossible for readers to compare mutations in the FL 1−3B (14;18)-positive group with the FL 3A/3B (14;18)-negative group. The authors should revise the figure, for example, by listing the variants in the FL 1−3B (14;18)-positive group in Figure 1A and the variants in the FL 3A/3B (14;18)-negative group in Figure 1B.
- Missense variants can be either pathogenic or benign. The authors should assess the pathogenicity of the variants and list all variants detected in this manuscript, which could be presented in the manuscript or as supplementary data.
- The rationale for selectively analyzing the “missense” variants in the KAT domain of the CREBBP gene is unclear. Why did the authors exclude other pathogenic variants (e.g. the truncating variants and the p.1680del variants)?
- The potential effect of all pathogenic variants (regardless of their presence in the KAT domain) on the diagnosis and prognosis of FL should be analyzed. If the authors have already analyzed all pathogenic variants and found no significance, they should describe their findings in the manuscript. This would support the rationale for focusing on the “missense” variants in the KAT domain in this study.
Minor:
Several typos were found, such as “FL 1-2/3A 14;18-positive” in line 134, “FL 3A/3B 14;18- group” in line 147, and “FL 3A-3B 14;18-positive” in line 184. These careless mistakes could lead to misunderstandings.
Author Response
Response to Reviewer 3 Comments
Dear Reviewer,
We appreciate your precious time in reviewing our paper and providing valuable comments. We have carefully considered the comments and tried our best to address every one of them. Our point-by-point responses and modifications in the manuscript have been highlighted in red.
Comments 1: In lines 144-150, the authors state, “In the samples of patients with FL 1−3B (14;18)-positive, missense mutations of the CREBBP gene in the KAT domain prevailed and were detected in 20/32 (63%) cases. CREBBP KAT missense mutation wasn’t detected in any case among patients with FL 3A/3B 14;18- group (Figure 1). In 12/32 (37%) patients from the FL 1−3B (14;18)-positive group and in 3/3 (100%) from the FL 3A/3B (14;18)- group, CREBBP gene mutations were represented by nonsense mutations, frameshift mutations, or multiple mutations (Figure 1).” However, Figure 1 only lists the variants identified in the study, making it impossible for readers to compare mutations in the FL 1−3B (14;18)-positive group with the FL 3A/3B (14;18)-negative group. The authors should revise the figure, for example, by listing the variants in the FL 1−3B (14;18)-positive group in Figure 1A and the variants in the FL 3A/3B (14;18)-negative group in Figure 1B.
Response 1: We completely agree that Figure 1 requires significant editing to make it more readable. We have provided a revised version.
Comments 2: Missense variants can be either pathogenic or benign. The authors should assess the pathogenicity of the variants and list all variants detected in this manuscript, which could be presented in the manuscript or as supplementary data.
Response 2: Thank you for pointing this out. The list of variants has been added as a supplementary data.
Comments 3: The rationale for selectively analyzing the “missense” variants in the KAT domain of the CREBBP gene is unclear. Why did the authors exclude other pathogenic variants (e.g. the truncating variants and the p.1680del variants)?
Response 3: Here, we relied on the findings published in the work of K Dreval et al. (DOI:10.1182/BLOOD.2022018719). They have shown that missense mutations in the KAT domain and other mutations (outside the KAT domain, multiple and truncating mutations in the KAT domain) might have different pathogenic effects. However we included the truncating variants and the p.1680del variant (outside the KAT domain) into analysis. Patients with these mutations belonged to the KATmiss- group in our study.
Comments 4: The potential effect of all pathogenic variants (regardless of their presence in the KAT domain) on the diagnosis and prognosis of FL should be analyzed. If the authors have already analyzed all pathogenic variants and found no significance, they should describe their findings in the manuscript. This would support the rationale for focusing on the “missense” variants in the KAT domain in this study.
Response 4: The patient with all types of mutations were included into analysis. However it was suggested previously by Dreval et al, that single missense mutations in the KAT domain can be used to approximate low-risk FL, while all other pathogenic variants are associated with an unfavorable prognosis and a short time to transformation. A description of this study is provided at lines 106-118. Therefore we attempted to test this hypothesis. We agree that this statement needs to be more clearly expressed in the text, and we have added а description of mutations at lines 145-156 and supplemental material with the list of detected mutations.
Comments 5 (minor): Several typos were found, such as “FL 1-2/3A 14;18-positive” in line 134, “FL 3A/3B 14;18- group” in line 147, and “FL 3A-3B 14;18-positive” in line 184. These careless mistakes could lead to misunderstandings.
Response 5: We agree with this comment and made corrections in text.
Round 2
Reviewer 3 Report
Comments and Suggestions for Authors
The most critical issues in the original manuscript have been addressed in this revised version. Although the current data may still be insufficient to support the conclusions claimed by the authors due to the low sample size, it could potentially contribute to further investigations of the disease.